# Applications of qualitative grounded theory methodology to investigate hearing loss: protocol for a qualitative systematic review

Yasmin H K Ali ,[1,2] Nicola Wright,[3] David Charnock,[3] Helen Henshaw ,[1,2] Derek Hoare [1,2]

¹Hearing Sciences, National Institute of Health Research Nottingham Biomedical Research Centre, Nottingham, UK
²School of Medicine, Clinical Neuroscience, Hearing Sciences, University of Nottingham, Nottingham, UK
³School of Health Sciences, Queen's Medical Centre, University of Nottingham, Nottingham, UK

**Correspondence to**
Yasmin H K Ali;
yasmin.ali@nottingham.ac.uk

## ABSTRACT

**Introduction** Hearing loss is a chronic condition affecting 12 million individuals in the UK. People with hearing loss regularly experience difficulties interacting in everyday conversations. These difficulties in communication can result in a person with hearing loss withdrawing from social situations and becoming isolated. While hearing loss research has largely deployed quantitative methods to investigate various aspects of the condition, qualitative research is becoming more widespread. Grounded theory is a specific qualitative methodology that has been used to establish novel theories on the experiences of living with hearing loss.

**Method and analysis** The aim of this systematic review is to establish how grounded theory has been applied to investigate the psychosocial aspects of hearing loss. Methods are reported according to the Preferred Reporting Items for Systematic Reviews and Meta-Analysis Protocols 2015 checklist. Studies included in this review will have applied grounded theory as an overarching methodology or have grounded theory embedded among other methodologies. Studies included will have adult participants (≥18 years) who are either people with an acquired hearing loss, their family and friends (communication partners), or healthcare practitioners including audiologists, general practitioners, ear, nose and throat specialists and hearing therapists. The quality of application of grounded theory in each study will be assessed using the Guideline for Reporting and Evaluating Grounded Theory Research Studies.

**Ethics and dissemination** As only secondary data will be used in this systematic review, ethical approval is not required. No other ethical issues are foreseen. This review is registered with the International Prospective Register of Systematic Reviews (http://www.crd.york.ac.uk/PROSPERO). Findings will be disseminated via peer-reviewed publications and at relevant academic conferences. Findings may also be published in relevant professional and third sector newsletters and magazines as appropriate. Data will inform future research and guideline development.

**PROSPERO registration number** CRD42019134197.

## Strengths and limitations of this study

► This systematic review is the first to provide a comprehensive critique of the use of grounded theory to investigate hearing loss.
► Methodological quality of studies is assessed using the Guideline for Reporting and Evaluating Grounded Theory Research Studies.
► Because experiences and articulations of hearing loss are influenced by age, only adult (≥18 years) participants (people with hearing loss, communication partners and healthcare practitioners) will be considered.
► The search will not include grey literature.

## INTRODUCTION

Hearing loss is a chronic condition that involves a decrease in an individual's ability to hear sounds. It can occur at mild, moderate, severe or profound levels, and typically worsens over time.[1] Twelve million people in the UK currently live with hearing loss.[2,3] This number is expected to increase to 15.6 million by 2035.[3] Globally, over 900 million people are expected to acquire a disabling hearing loss by 2050.[2,3]

The annual global societal cost of unaddressed hearing loss is $750 billion[4]. Within the UK, the annual cost of hearing loss is £30 billion, with the larger proportion of this cost dealing with the social, psychological and health impacts of hearing loss, rather than the treatments provided by audiology services.[5] Untreated hearing loss can impact the work opportunities and occupational progression of people with hearing loss (PHL).[4] Estimations of the global occupational limitations occurring due to the stigma and inequalities associated with hearing loss are $105 billion annually.[4]

PHL experience functional limitations such as communication difficulties in

day-to-day conversations, where speech is often misheard and becomes challenging to follow.[6–8] Failed instances of communication can lead PHL to experience embarrassment, for example, if incorrect responses are given in a conversation after mishearing others.[9 10] Because PHL struggle to have full interactions with others, they can come to feel isolated and separated from the world.[10–12] The anxiousness of repeatedly not being able to hear others can lead to withdrawal from social situations.[13] Consequently, it is estimated that PHL are more likely to develop depression than the general population.[14–17]

The impact of hearing loss is not only psychological, as the interpersonal relationships of PHL are also affected. Not being able to fully communicate and establish successful interactions can also lead to feelings of frustration.[18] Individuals that communicate with PHL, such as their spouses, family and friends, can also become frustrated due to not being heard or understood in interactions.[18 19] Conflict in the relationships of PHL and their communication partners have been reported to occur,[18 20 21] with hearing loss being a significant risk factor for the development of abusive relationships.[22]

Investigations into the experience of hearing loss are crucial for understanding the implications it has for the person, and their care. To date, most studies in hearing loss research have used quantitative methods. Due to the evident psychosocial impacts of hearing loss, there has been a significant increase in the adoption of qualitative methodologies in hearing loss research,[23] particularly since the publication of 'Conducting qualitative research in audiology: A tutorial' in 2012 . Qualitative methods allow researchers to understand the experiences, opinions and perspectives associated with hearing loss, which experimental measures are not designed to uncover.[23–25]

There are five main qualitative methodologies; phenomenology, ethnography and narratives which mainly focus on investigating descriptive characteristics of a phenomenon, and case studies and grounded theory which facilitate both descriptive and explanatory understandings of a phenomenon.[26] Grounded theory is a qualitative methodology developed specifically to enable the construction of novel theories that are directly emergent from within the data, which comprehensively explore and explain phenomena.[26 27]

Grounded theory was first established by Glaser and Strauss in 1967 .[28] It is defined as the systematic exploration of data in an open-minded, comparative and rigorous manner for developing a novel theory that is purely grounded within the data.[27 29] Through applying grounded theory, people's experiences and occurring social processes can be understood by integrating definitions and meanings from the perspectives of individuals from the target population.[30 31] To do this, a researcher can adopt the principles of one of three grounded theory schools (table 1).

Assessing the methodological applications of grounded theory is essential for determining the trustworthiness and credibility of the theories developed, as suggested by researchers in various fields such as psychology,[32] nursing,[33] physiology[34 35] and business and management.[36] A field that has applied and critiqued applications of grounded theory is chronic illness research.[37–40] The founders of all three schools of grounded theory (table 1) also specialise in investigating chronic illnesses. Glaser *et al* initially formed the methodology to investigate fatal chronic illnesses,[28 29] but then used it more extensively to investigate non-fatal chronic conditions.[38 41 42] Charmaz has since identified the methodology as the most appropriate for establishing novel knowledge regarding life-long conditions.[27 38 42–44] Hearing loss is also a chronic disease as recognised by the WHO,[45 46] and is the third most common chronic condition affecting the

**Table 1** The three different schools of grounded theory methodology

| Dimensions of comparison | Glaserian school (1967)[28 29] | Straussain school (1990)[31] | Constructivist school (2006)[27] |
|---|---|---|---|
| Philosophical stance | Empiricism | Interpretivism | Constructivism |
| School founders | Barney Glaser and Anselm Strauss | Anselm Strauss and Juliet Corbin | Kathy Charmaz |
| Philosophical principle | Knowledge is formed based on the individual experience and can be objectively measured. | Knowledge is subjective and socially constructed. Establishing one 'truth' is impossible. | Knowledge is subjective and is constructed by both the participant's experiences and the researcher's own interpretations. |
| Researcher's influence | Researchers can completely detach themselves from their research and not influence it. | Researchers cannot detach themselves from the research and will always influence the research process and findings. | A comprehensive truth is pursued; however, it will only be reflective of the social context and group being studied. |
| Grounded theory emphasis | Constant comparative analysis: constantly comparing data and outcomes to establish objective knowledge. | Reflexivity: researcher provides reflections on the process of data collection and analysis, and recognises how they influence this process. | Constant comparative analysis, reflexivity, theoretical sampling (knowledge is pursued and collected through recruiting different samples and investigating different concepts to develop the theory). |

population.[47–49] Despite this, none of the aforementioned research into chronic illness reviewing grounded theory included hearing loss as a condition for consideration.

There has been an increase in the application of grounded theory methodology in the field of hearing loss.[50] A lack of consistency in the use of grounded theory across healthcare research has been reported[51] despite core principles being maintained across its three schools, and the emphasis of grounded theory being a rigorous systematic process.[27 29 50–52] The literature reinforces the importance of avoiding misinformed applications of grounded theory when investigating hearing loss,[24 50] to ensure that theories developed are sufficiently trustworthy for informing subsequent studies.[24 50 51 53] Therefore, this systematic review has outlined grounded theory as a qualitative methodology needed for review regarding its applications in hearing loss research.

No methodological systematic review has yet been conducted to assess the quality of studies that use grounded theory to investigate hearing loss. This is essential to inform future applications of grounded theory for improved and trustworthy hearing loss research. Therefore, this review primarily aims to (1) critically appraise the execution of grounded theory methodology in hearing loss research in terms of methodological quality and (2) produce recommendations for future grounded theory methodological applications pertaining to hearing loss research. A secondary aim is to (3) describe how grounded theory methodology has been applied within the field of hearing loss.

## METHODS AND ANALYSIS

The methods and analysis of this systematic review will be reported in accordance to the Preferred Reporting Items for Systematic Reviews and Meta-Analysis (PRISMA) 2015 checklist.[54 55] The recommended items on the PRISMA Protocols checklist will form the subheadings of this section. This systematic review will also follow the Cochrane Handbook's suggested approach for conducting methodology-based systematic reviews, that is, Studies, Data, Methods, and Outcomes (SDMO) structure.[56 57] The SDMO is used to investigate contemporary research methods to illuminate the impact of the methodology on the quality of research within a specific field.[56]

### Timeframe

Initial searches were conducted in June 2019 but will be updated 1 month prior to submission of the manuscript for review.

## ELIGIBILITY CRITERIA
### Types of studies

Studies included in this review will have used grounded theory methodology with an appropriate peer-reviewed reference for that adopted approach. For a study to be included, it can apply grounded theory as an overarching methodology, or be embedded into it among other qualitative methodologies such as case studies, ethnography, narratives and phenomenology. During full-text screening, if a study explicitly states that grounded theory was used during data collection and/or analysis, then they will be eligible for inclusion. Studies that do not explicitly state use of grounded theory methodology will be excluded. Purely quantitative studies, studies that have not applied grounded theory methodology, articles reporting expert opinions, case reports, practice guidelines, case series, conference abstracts and book chapters will be excluded. Only studies published in English will be included. Any studies published prior to 1967 (when grounded theory methodology was first introduced to the literature; Glaser et al[28]) will be excluded.

### Types of data

Original research submitted to peer-reviewed journals will be the data used in this systematic review. The data collected in those included studies will include qualitative primary data occurring in the form of transcripts or quotes from audio interviews with participants, patient journals, written reflections of patients, family members of PHL, or professionals working within audiology, memos of progression of study, initial themes and analyses, and observational notes. Records reporting diary entries of patients before participation in their prospective study will also be eligible for inclusion. No secondary data will be collected.

### Types of methods

The methodology under review in this systematic review is qualitative grounded theory methodology. Different methods, such as interviews, observations and focus groups, can all be undertaken using grounded theory methodology. Based on the principles of grounded theory and the school of grounded theory followed, the most appropriate methods are selected. The methods used in each study will be identified and discussed.

### Types of outcomes

The outcomes will be extracts taken verbatim from included articles. These extracts will detail the specific steps and decisions made in using grounded theory methodology and how it was applied in the study. Data extracted will include participant characteristics, data collection methods, particular type of hearing loss being investigated, school of grounded theory followed if mentioned, attempts to establish qualitative rigour or trustworthiness, study/methodology limitations, advantages and disadvantages of using grounded theory if explicated by authors, recommendations, among other things. These data will then be critically appraised using the Guideline for Reporting and Evaluating Grounded Theory Research Studies (GUREGT) guideline for assessing the application of grounded theory methodology in research.

### Settings

Any research setting will be included.

## Participants

Participants will include people with mild to profound, acquired or congenital, hearing loss. Studies involving individuals with no residual hearing will be excluded. This is because people with no residual hearing often identify differently (and have very different experiences) to people with some residual hearing.[58–60] Studies involving both deaf and hearing loss participants may ask each individual which group they identify with more and classify them on that basis. These studies will be eligible for inclusion as distinctions between the two groups can be made. Studies that do not specify which participants are deaf and which have hearing loss will be excluded.

Only studies including adults (≥18 years) will be included. The levels of hearing loss for eligibility are mild, moderate, severe or profound, according to the British Society of Audiology.[61] Cochlear implant users, hearing aid users and non-hearing aid users will be included. Studies involving communication partners (friends, family members and colleagues that interact with PHL) will also be eligible for inclusion. Studies including healthcare practitioners such as audiologists, GPs, ENT specialists and hearing therapists involved in hearing loss treatment will be considered. Studies with children and young adolescents (<18 years) will be excluded, unless they are in a study with adults where the adult data are reported separately.

## Information sources

A systematic search strategy will be employed to identify peer-reviewed journal articles that meet the eligibility criteria. The following databases will be used: PsycINFO (1800s–current), Applied Social Sciences Index and Abstracts (1987–current), Global Health (OvidSP database, 1973–current). Web of Science (1899–current), PubMed (1996–current), British Nursing index (1994–current), Cumulative Index to Nursing and Allied Health Literature (1961–current), MEDLINE (Ovid, In-Process & Other Non-Indexed Citations, 1946–current), Scopus (1983–current) and EBSCO (1944–current). Google Scholar will be used for forward citation tracking.

Additionally, snowballing of the reference lists of articles shortlisted for inclusion will be undertaken, and related articles from shortlisted authors will be screened in an attempt to identify any relevant articles which may not have been returned by the database searches. The electronic database searches will be updated just before the final analyses and any eligible studies retrieved will be included. At the end of the study selection process, the search strategy produced for each database will be reported, in addition to a PRISMA flow diagram.

To check the feasibility of this systematic review, a scoping search was conducted across multiple databases including Medline, PubMed and PsycINFO. This indicated that there were sufficient relevant papers, approximately 30, to perform a useful synthesis. The searches also informed the key terms for use in the formal search strategy.

## Search strategy

The search strategy was formed from free text and controlled vocabularies (Medical Subject Headings), consultations with an information specialist at the University of Nottingham, and the testing of search results.

1. Hearing Loss, Noise-Induced/ or Hearing Loss/ or Hearing Loss, Sensorineural/ or Hearing Loss, Unilateral/ or Hearing Disorders/ or Hearing Loss, Sudden/ or Hearing Loss, Bilateral/ or Hearing Tests/ or Hearing Loss, Conductive/
2. hearing loss*.mp. or exp Hearing Loss/
3. 1 or 2
4. Amplifiers, Electronic/ or amplifier*.mp.
5. Hearing Aids/ or listening device*.mp.
6. Cochlear Implants/ or cochlear implant*.mp. or Cochlear Implantation/
7. exp Hearing Aids/ or hearing aid*.mp.
8. 4 or 5 or 6 or 7
9. hard of hearing*.mp.
10. communication partner*.mp.
11. audiologist*.mp. or Audiologists/ or Audiology/
12. exp Persons With Hearing Impairments/
13. 9 or 10 or 11 or 12
14. grounded theory*.mp.
15. exp Grounded Theor*/
16. 14 or 15
17. 3 or 8 or 13
18. 16 and 17

## Data management

YHKA will be responsible for management of the data. A digital record will be kept for all identified articles, and the process of data screening and extraction will be tracked in Covidence. A unique ID code will be assigned to the included articles for access to the full text and data collection sheet.

## Selection process

This systematic review will follow the Centre for Reviews and Dissemination guidance for undertaking reviews in healthcare[62] and suggested systematic review practice. All data screening and extraction phases will be independently completed by at least two researchers.[62 63] Any differences will be resolved through discussion or by consulting a third author. Studies that do not meet the inclusion criteria will be excluded from the review and the reason for their exclusion at full-text stage will be reported. For a summary of the data selection, extraction and analysis processes, please refer to figure 1.

## Data extraction process

After piloting the data extraction form by an independent author, YHKA will extract data for all studies and 100% of the studies will be extracted by a second researcher(s) in line with published guidance.[62] Any disagreements that cannot be resolved through discussion will be resolved by a third author. Extracted data will be in the form of direct text collected from the included articles. Data will

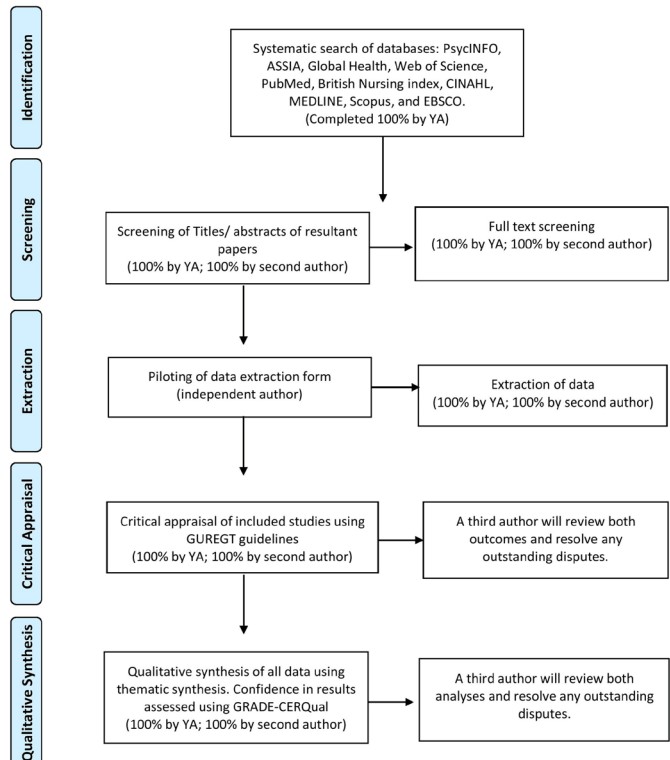

Identification

Screening

Extraction

Critical Appraisal

Qualitative Synthesis

Systematic search of databases: PsycINFO, ASSIA, Global Health, Web of Science, PubMed, British Nursing index, CINAHL, MEDLINE, Scopus, and EBSCO.
(Completed 100% by YA)

Screening of Titles/ abstracts of resultant papers
(100% by YA; 100% by second author)

Full text screening
(100% by YA; 100% by second author)

Piloting of data extraction form
(independent author)

Extraction of data
(100% by YA; 100% by second author)

Critical appraisal of included studies using GUREGT guidelines
(100% by YA; 100% by second author)

A third author will review both outcomes and resolve any outstanding disputes.

Qualitative synthesis of all data using thematic synthesis. Confidence in results assessed using GRADE-CERQual
(100% by YA; 100% by second author)

A third author will review both analyses and resolve any outstanding disputes.

**Figure 1** Flowchart outlining the systematic review process. CERQual, Confidence in the Evidence from Reviews of Qualitative research; GUREGT, Guideline for Reporting and Evaluating Grounded Theory Research Studies.

be stored in Covidence. Prior to starting data extraction guidance notes will be formed by YHKA.

Extracted data will include:

1. The study's title, authors, date of publication, number of citations and country.
2. Date(s)/time period of data collection.
3. Aims, objectives and/or research questions.
4. Source(s) of funding.
5. Statement of conflict(s) of interest.
6. Participant characteristics: for example, age, sample size, gender, country and occupation status.
7. Key participant characteristics: type and severity of hearing loss, hearing aid usage and whether they have a cochlear implantation.
8. Type of participants included (PHL, their family/friends—communication partners, healthcare practitioners such as audiologists, GPs, ENT specialists and hearing therapists).
9. Methods of recruitment of participants.
10. Data collection methods.
11. Particular type of experience of hearing loss being investigated.
12. School of grounded theory followed if mentioned.
13. Ethical standards/approval.
14. Attempts to establish qualitative rigour or trustworthiness.
15. Study/methodology limitations.
16. Advantages and disadvantages of using grounded theory if explicated by authors.
17. Study design: methodology and methods.
18. Key findings.
19. Conclusions.
20. Recommendations.

## Methodological quality of the individual studies

Quality appraisal of studies will be conducted using the GUREGT,[64] and critiques of the methodological quality of each individual study will be established. YHKA will appraise all records, and 100% of the studies included will be independently appraised by a second author using the same guidelines. A third reviewer will review both critique outcomes and resolve any outstanding disputes.

## Data synthesis

A thematic synthesis approach as established by Thomas and Harden[65] will be used to synthesise the findings. Through this approach, four stages will be undertaken to achieve codes, refine themes and synthesise the data. These four stages are (1) familiarisation with data through in-depth reading of included papers, (2) line by line coding of the extracted data and critical appraisal results, (3) categorising these codes into descriptive themes and (4) developing these descriptive themes into analytical ones which provide deeper insights than the original studies.[66]

This process will consist of coding the data extracted from the included articles and applying initial codes which are close to the analysed data. This will involve establishing which aspects of hearing loss were investigated, how they were investigated and how grounded theory was used as a methodology in its applications of central principles such as theoretical sampling, constant comparative analysis and theoretical saturation. for investigating the specific hearing loss area. The results of the critical appraisal will then illustrate where grounded theory has been used rigorously and less so, to investigate hearing loss. From descriptive themes, analytical themes will follow, and these will inform a set of guiding statements for researchers wishing to use grounded theory to investigate hearing loss in future studies. Overall, thematic synthesis was chosen as it enables the greater generalisability of results.

The data synthesis will be 100% completed by YHKA and 100% by a second author, ensuring inter-rater reliability as required for qualitative rigour.[67 68] The results will be compared and discussed in meetings between both authors. A third author will also independently compare both analyses, provide feedback and solve any outstanding disputes. The GRADE-CERQual ('Confidence in the Evidence from Reviews of Qualitative research')[69] tool will be used to assess the confidence of the data synthesis findings.

## Risk of bias

Within the research team, two are the qualitative experts (NW and DC), with the remaining four members having experience of conducting qualitative research investigating aspects of hearing loss (HH, DH and YHKA). DC and YHKA have experience in the application of grounded theory methodologies. To avoid bias during the conduct of this systematic review, each author will reflect on their theoretical predispositions, biases, decisions made during the research process and any problems faced. The risk of bias during analysis will be avoided by assessing studies using the GUREGT guidelines, the GRADE-CERQual tool to assess confidence in the data synthesis, and by considering the funding and conflict(s) of interest statements outlined by each study during data extraction to evaluate whether selective reporting of outcomes was present.

## Ethics and dissemination

As only secondary data will be used in this systematic review, no ethical approval is required. No other ethical issues are foreseen. The International Prospective Register of Systematic Reviews (http://www.crd.york.ac.uk/PROSPERO) holds the registration record of this systematic review.

The results from the systematic review will be disseminated via peer-reviewed publications and relevant academic conferences. Findings may also be published in relevant professional and third sector newsletters and magazines as appropriate. Data will be used to inform future research and guideline development.

## Outcome of systematic review

The primary outcome of this review will be a comprehensive understanding of how grounded theory methodology has been used to investigate hearing loss. This will be presented in a detailed table displaying the main themes and subthemes reached after the critical appraisal process using GUREGT and the qualitative thematic synthesis of data. Another primary outcome will be a set of recommendations highlighting the main downfalls of studies that have investigated hearing loss using grounded theory, and guidance for future researchers on how to apply the methodology most effectively. These will be presented in a finalised list based on the main themes and subthemes established through critical appraisal and thematic synthesis. A secondary outcome will be a table summarising the topics of hearing loss that were investigated using grounded theory, as established from data extraction.

## Summary

This systematic review will be the first to establish how grounded theory has been used to investigate hearing loss. A critical appraisal of all hearing loss studies that have investigated hearing loss within adult populations using grounded theory will be performed. This novel systematic review will aid in implementing more precise applications of grounded theory in future research. This will also enable more refined understandings of the experiences of hearing loss to be established and facilitate better care for those who have hearing loss.

**Acknowledgements** Thanks to Dr Melanie Ferguson for comments on an earlier version of this protocol.

**Contributors** YHKA is the guarantor of the review (CRD42019134197). YHKA led on the development of the review protocol and drafted the manuscript. YHKA, NW, DC, HH and DH contributed to the development of the eligibility criteria and selection process. NW, DC, HH and DH all read drafts of the manuscript, provided feedback and approved the final manuscript.

**Funding** This systematic review presents independent research with differing sources of funding. YHKA is funded by the National Institute for Health Research (NIHR) Nottingham Biomedical Research Programme and Sonova Holding AG. DH is funded by the NIHR Biomedical Research Centre Programme. HH is funded through an NIHR Career Development Fellowship (NIHR Ref: CDF-2018–11-ST2-016). NW and DC are funded by the University of Nottingham.

**Disclaimer** The views expressed are those of the authors and not necessarily those of the NHS, the NIHR or the Department of Health and Social Care.

**Competing interests** None declared.

**Patient and public involvement** PHL were consulted in the identification of the most relevant healthcare practitioner roles when seeking hearing healthcare services. The prominent professions identified were audiologists, general practitioners (GPs), ear, nose and throat (ENT) specialists and hearing therapists. Therefore, they have been included within the inclusion criteria. Patient and public involvement will also include the preparation and dissemination of a lay summary of the review findings for a general audience. This will be achieved with appropriate training and support, provided by the NIHR Nottingham Biomedical Research Centre Patient and Public Involvement (PPI) manager.

**Patient consent for publication** Not required.

**Provenance and peer review** Not commissioned; externally peer reviewed.

**ORCID iDs**
Yasmin H K Ali http://orcid.org/0000-0002-0228-6562
Helen Henshaw http://orcid.org/0000-0002-0547-4403
Derek Hoare http://orcid.org/0000-0002-8768-1392

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
