## [Reviewer comments · BMJ Open]

ARTICLE DETAILS

TITLE (PROVISIONAL)	Applications of qualitative grounded theory methodology to investigate hearing loss: Protocol for a qualitative systematic review
AUTHORS	Ali, Yasmin; Wright, Nicola; Charnock, David; Henshaw, Helen; Hoare, Derek

VERSION 1 – REVIEW

REVIEWER	Katherine Allan St. Michael's Hospital, Toronto ON Canada
REVIEW RETURNED	20-Aug-2019

GENERAL COMMENTS	Thank you for the opportunity to review this systematic review protocol that aims to establish how grounded theory has been applied to investigate the psychosocial aspects of hearing loss. The protocol is well written, however I have some concerns about a few methodological limitations. 1. The strengths and limitations need to be more broad and not related to your methods. Readers don't want to know if you developed your search strategy with an information specialist, as this is pretty much a given of all systematic reviews. Same goes for using the PRISMA-P guidelines. Readers want to know why is this study important? (i.e. why should I care?) or what makes your study unique? Please revise the strengths according to why your study is important and needed.2. The types of outcomes could be defined more explicitly - I find them very general and so am not entirely sure what you plan to collect or analyze.3. I am most concerned about the data management process, including selection, data extraction, assessment of quality and inter-rater reliability of coding. You mention that only 20% of all studies will be reviewed by a second author for all of these stages. It is strongly recommended in systematic reviews to have all of these stages performed independently by at least 2 reviewers, with the use of a 3rd reviewer for resolution of disagreements. I strongly suggest that you change your planned methodology to reflect this practice, as potential reviewers of the manuscript will comment and possibly reject based on poor execution/methodology.
--

REVIEWER	Dr. Maximilian Sohn Klinik für Allgemein-, Viszeral-, Endokrine- und Minimalinvasive Chirurgie, München Klinik Bogenhausen, Munich City Hospital, Germany
-----------------	--

REVIEW RETURNED	23-Aug-2019
-------------

GENERAL COMMENTS	This is a well written protocol of an systematic review to assess how grounded theory has been used to investigate hearing loss. All relevant requirements of the PRISMA-P Statement and the Cochrane guidelines for systematic reviews are incorporated. The methodology is comprehensive and adequate. Appropriate quality assessment is applied. Please add the timeframe of the study. When will the literature review be started and completed. When will data extraction, evaluation and synthesis be done and when is the publication of the manuscript planned.
--

REVIEWER	Nikki Rousseau University of Leeds, United Kingdom
-----------------	---

REVIEW RETURNED	30-Oct-2019
-------------

GENERAL COMMENTS	Although I have responded "no" in several places on the review checklist, in all cases a better response would have been "mostly" and I've indicated "no" mainly to give the authors a sense of where their protocol would benefit from some further elaboration or clarification. Although it is clear that the aim of this review is to identify and synthesise research which has used a grounded theory approach to understand experiences of hearing loss in adults, what was less clear to me was why this focus had been chosen, and to exactly what body of research it will add. If the aim is to understand experiences of hearing loss - then why grounded theory papers only? (rather than other qualitative research). I assume that there is some methodological aim around grounded theory but this needs to be clearer – and again – what is the research gap that this will address, and for who and how will the findings be helpful. Without this it is a little hard to judge whether the methods chosen are appropriate for the study aims, and I think the authors will struggle as they conduct their review to know what to include, what data to extract etc. The authors might find it helpful to identify a primary and secondary aim - the primary aim might be around understanding and critiquing the application of grounded theory methods and the secondary aim might be to synthesise the hearing loss findings (or it might be the other way round). Eligibility criteria – types of studies. This section currently lists a range of quantitative study types but doesn't list qualitative study types – this is a little confusing. The authors do not justify their choice of thematic synthesis. This seems particularly like an omission given that the focus of the review is on a specific (and different) qualitative approach. They also cite two references by Braun and Clarke. Although I'm less familiar with these two references than with Braun and Clarke 2006, it looks like these references are for thematic analysis rather than synthesis. Whilst a thematic analysis could be undertaken of these data, there are other more commonly cited references for thematic synthesis (e.g. Thomas J , Harden A . Methods for the thematic synthesis of qualitative research in systematic reviews. BMC Med Res Methodol 2008;8:45.doi:10.1186/1471-2288-8-45). I'd like it be clearer exactly what and how the authors are doing in terms of synthesis and to see a justification for the choice of synthesis approach.
--

	Also given the focus it would have been helpful to have understood the background of those doing the research, and perhaps to have seen them be reflexive about their positions. There are a number of aspects of the planned review where I am unclear about exactly what will and will not be included. The authors make no reference to a scoping search and I wondered whether they have done this and got a sense of how many studies they are potentially looking at and where the issues might be in terms of inclusion/exclusion.  • I'm not sure what the "grounded theory" keywords encompass but wondered whether additional textword searches e.g. "constant comparison" "theoretical sampling" might identify research that might otherwise be missed. Or look for papers which cite key GT texts? • I wasn't sure what the limits are for the research in grounded theory. There are many people who take aspects of grounded theory as part of their qualitative approach but who do not "claim" to be using grounded theory. I conducted a review of qualitative research in ENT and found that the underlying approach was often unclear, particularly when reporting in journals with a shorter word limit or where the audience might be less familiar with different qualitative approaches. • The authors say that "The data collected in those included studies will include qualitative primary data occurring in the form of transcripts or quotes from audio interviews with participants, patient journals, written reflections of patients, family members of PHL, or audiologists, memos of progression of study, initial themes and analysis, and observational" (this sentence appears not to be complete). Here, and elsewhere, the authors only refer to audiologists rather than other health professionals – I wasn't sure whether the authors are interested in other health professionals who are involved in the care of people with hearing loss and their perspectives and interactions with people with hearing loss as well? • There is some variation in terminology for hearing loss. Terminology from mild to profound is used in some places but elsewhere the authors refer to "people with no residual hearing" (as an exclusion criteria). Their argument is that the experience of total deafness is different from that of hearing loss e.g. in terms of identity – this may be true but it is likely that there is a continuum with the experience of profound hearing loss but some hearing very similar to that of having no residual hearing at all, (and that other factors are also relevant such as age at which hearing was lost and whether other people in their family have hearing loss). This is partly about how the authors will handle exclusion criteria in practice (if papers refer to profound hearing loss for example and don't explicitly refer to whether residual hearing is present) but this is also somewhere that PPI might have been helpful in helping the authors to think through and justify their focus. This is also somewhere that a scoping search might have been helpful. • Linked to the previous point – what will the authors do where the sample is mixed and includes some people with no residual hearing? • I couldn't see anything about exclusion criteria around language. Or about years searched (the authors could exclude anything published prior to 1967) The authors say that "A review of all titles and abstract resulting from the searches will be performed by YA. During this screening stage, 20% of titles/abstracts will be independently reviewed by a
--	---

	second author. Any differences will be resolved by a third author. Full text screening will be carried out by YA and 20% of full texts will be independently screened by a second author. Any differences between the two authors will be resolved by a third author” It is unclear exactly what (how will the 20% be selected?), when, and why these 20% samples are being undertaken. It might be best to do at least some of this early on in each of the processes to identify any areas of discrepancy, resolve them, and make adjustments to e.g. the inclusion criteria – if the authors wait until they are half way through to carry out this activity then if a problem is identified then a large amount of re-screening would need to be done. I’m not sure that doing methodology research is reason enough not to have a PPI element. Patients might have a view on what the boundaries of the research should be (e.g. around which health professionals are important in supporting them with their hearing loss) or a view on data that it would be interesting to extract (although the focus is on grounded theory the authors do plan to synthesise data on experiences of hearing loss and these findings may be of interest to patients). Patient groups might also be able to help with dissemination. It doesn't need to be a large element.
--	---

VERSION 1 – AUTHOR RESPONSE

Reviewer’s Feedback

Reviewer 1:

Comment 1: “The strengths and limitations need to be more broad and not related to your methods. Readers don't want to know if you developed your search strategy with an information specialist, as this is pretty much a given of all systematic reviews. Same goes for using the PRISMA-P guidelines. Readers want to know why is this study important? (i.e. why should I care?) or what makes your study unique? Please revise the strengths according to why your study is important and needed.”

Thank you for your comment. Whilst we agree readers want to know why the study is important, as per BMJ Open style and as confirmed by the editor, it is a requirement that the strengths and limitations should relate specifically to the methods and design of the study. As such, we have clearly stated this in the introduction section.

“There has been an increase in the application of grounded theory methodology in the field of hearing loss¹. A lack of consistency in the use of grounded theory across healthcare research has been reported² despite core principles being maintained across its three schools, and the emphasis of grounded theory being a rigorous systematic process¹⁻⁵. The literature reinforces the importance of avoiding misinformed applications of grounded theory when investigating hearing loss^{1,6}, to ensure that theories developed are sufficiently trustworthy for informing subsequent studies^{1,2,6,7}.

Therefore, this systematic review has outlined grounded theory as a qualitative methodology needed for review regarding its applications in hearing loss research.

No methodological systematic review has yet been conducted to assess the quality of studies that use grounded theory to investigate hearing loss. This is essential to inform future applications of grounded theory for improved and trustworthy hearing loss research.”

Comment 2: “The types of outcomes could be defined more explicitly - I find them very general and so am not entirely sure what you plan to collect or analyze.”

Paragraph in question: “The explicit use of grounded theory methodology as outlined in the methods section of studies investigating hearing loss. Where available, specific mentions of the applications of grounded theory and the type of school followed will be extracted. Studies must also have researched topics regarding hearing loss with grounded theory methodology, and these topics will also be extracted and collated.”

We have revised this section to be more specific. It now reads:

“The outcomes will be extracts taken verbatim from included articles. These extracts will detail the specific steps and decisions made in using grounded theory methodology and how it was applied in the study. Data extracted will include participant characteristics, data collection methods, particular type of hearing loss being investigated, school of grounded theory followed if mentioned, attempts to establish qualitative rigour or trustworthiness, study/ methodology limitations, advantages and disadvantages of using grounded theory if explicated by authors, recommendations, amongst other things. These data will then be critically appraised using the GUREGT guideline for assessing the application of grounded theory methodology in research.”

Comment 3: “I am most concerned about the data management process, including selection, data extraction, and assessment of quality. And inter-rater reliability of coding. You mention that only 20% of all studies will be reviewed by a second author for all of these stages. It is strongly recommended in systematic reviews to have all of these stages performed independently by at least 2 reviewers, with the use of a 3rd reviewer for resolution of disagreements. I strongly suggest that you change your planned methodology to reflect this practice, as potential reviewers of the manuscript will comment and possibly reject based on poor execution/methodology.”

Thank you very much for outlining this point. Now the manuscript clearly states that all data screening and extraction phases will be independently completed by at least two researchers^{62,63}. Any differences will be resolved through discussion or by consulting a third author. Studies that do not meet the inclusion criteria will be excluded from the review and the reason for their exclusion at full text stage will be reported. The “Selection Process” section now reads:

“This systematic review will follow the Centre for Reviews and Dissemination (CRD) guidance for undertaking reviews in healthcare⁶² and suggested systematic review practise. All data screening and extraction phases will be independently completed by at least two researchers^{62,63}. Any differences will be resolved through discussion or by consulting a third author. Studies that do not meet the inclusion criteria will be excluded from the review and the reason for their exclusion at full text stage will be reported.”

We have also added the following in the data extraction phase:

“After piloting the data extraction form by an independent author, YA will extract data for all studies and this will be checked for accuracy and completeness by a second author, in line with published guidance⁶². Any disagreements that cannot be resolved through discussion will be resolved by a third author.”

The following was added to the methodological quality of the individual studies section:

“Quality appraisal of studies will be conducted using the Guideline for Reporting and Evaluating Grounded Theory Research Studies (GUREGT)⁶⁵, and critiques of the methodological quality of each individual study will be established. YA will appraise all records, and 100% of the studies included will be independently appraised by a second author using the same guidelines. A third reviewer will review both critique outcomes and resolve any outstanding disputes.”

We hope that this reflects the quality of thoroughness of this systematic review.

Reviewer: 2

Comment 4: Please add the timeframe of the study. When will the literature review be started and completed. When will data extraction, evaluation and synthesis be done and when is the publication of the manuscript planned.

We have now added an expected time frame for the review in the methods section. Please refer to the Editor's fifth request for the added timeframe.

Reviewer: 3

Comment 5: "Although it is clear that the aim of this review is to identify and synthesise research which has used a grounded theory approach to understand experiences of hearing loss in adults, what was less clear to me was why this focus had been chosen, and to exactly what body of research it will add. If the aim is to understand experiences of hearing loss - then why grounded theory papers only? (rather than other qualitative research). I assume that there is some methodological aim around grounded theory but this needs to be clearer – and again – what is the research gap that this will address, and for who and how will the findings be helpful." "And to exactly what body of research it will add. If the aim is to understand experiences of hearing loss - then why grounded theory papers only? (rather than other qualitative research)."

The focus of this review is the use of grounded theory, specifically within the field of hearing loss research¹. The particular focus is on grounded theory because it is the only qualitative methodology that enables researchers to establish novel theories completely grounded within the data compared to the other four qualitative methodologies. The unique capacity offered by the methodology has seen it increasingly used within hearing loss research. We consider an authoritative systematic review is required to examine the use of grounded theory and its associated methodological rigour within the field, to inform future use in this field. The following sections have been edited in the introduction: "There are five main qualitative methodologies; phenomenology, ethnography, and narratives which mainly focus on acquiring descriptive characteristics of a phenomenon, and case studies and grounded theory which facilitate both descriptive and explanatory understandings of a phenomenon⁹. Grounded theory is the only qualitative methodology that enables the construction of novel theories that are completely emergent from within the data, and comprehensively explore and explain phenomena^{5,9}."

"There has been an increase in the application of grounded theory methodology in the field of hearing loss¹. A lack of consistency in the use of grounded theory across healthcare research has been reported² despite core principles being maintained across its three schools, and the emphasis of grounded theory being a rigorous systematic process¹⁻⁵. The literature reinforces the importance of avoiding misinformed applications of grounded theory when investigating hearing loss^{1,6}, to ensure that theories developed are sufficiently trustworthy for informing subsequent studies^{1,2,6,7}. Therefore, this systematic review has outlined grounded theory as a qualitative methodology needed for review regarding its applications in hearing loss research. No methodological systematic review has yet been conducted to assess the quality of studies that use grounded theory to investigate hearing loss. This is essential to inform future applications of grounded theory for improved and trustworthy hearing loss research. Therefore, this review primarily aims to (1) critically appraise the execution of grounded theory methodology in hearing loss research in terms of methodological quality, and (2) produce recommendations for future grounded theory methodological applications pertaining to hearing loss research. A secondary aim is to (3) describe how grounded theory methodology has been applied within the field of hearing loss."

Comment 6: "The authors might find it helpful to identify a primary and secondary aim - the primary aim might be around understanding and critiquing the application of grounded theory methods and the secondary aim might be to synthesise the hearing loss findings (or it might be the other way round)."

Thank you for raising this consideration. This review is primarily concerned with the use of grounded theory methodology within the field of hearing loss research. For clarification we have separated the aims into primary and secondary ones. Please see the final paragraph in response to comment 5.

Comment 7: "Eligibility criteria – types of studies. This section currently lists a range of quantitative study types but doesn't list qualitative study types – this is a little confusing"

Indeed, qualitative study types should also be listed here. We have revised the text as follows:
“ELIGIBILITY CRITERIA: Studies included in this review will have used grounded theory methodology with an appropriate peer-reviewed reference for that adopted approach. For a study to be included, it can apply grounded theory as an overarching methodology, or be embedded into it amongst other qualitative methodologies such as case studies, ethnography, narratives, and phenomenology. During full text screening, if a study explicitly states that grounded theory was used during data collection and/ or analysis, then they will be eligible for inclusion. Studies that do not explicitly state use of grounded theory methodology will be excluded. Included studies can be qualitative or mixed methods research of any design/ methodology; retrospective or prospective studies, before and after comparison studies, RCTs, non-RCTs, cohort studies, prospective observational studies, case-control studies, cross-sectional studies, longitudinal studies. Purely quantitative studies, studies that have not applied grounded theory methodology, articles reporting expert opinions, case reports, practice guidelines, case series, conference abstracts, and book chapters will be excluded.”

Comment 8: “The authors do not justify their choice of thematic synthesis. This seems particularly like an omission given that the focus of the review is on a specific (and different) qualitative approach. They also cite two references by Braun and Clarke. Although I’m less familiar with these two references than with Braun and Clarke 2006, it looks like these references are for thematic analysis rather than synthesis. Whilst a thematic analysis could be undertaken of these data, there are other more commonly cited references for thematic synthesis (e.g. Thomas J , Harden A . Methods for the thematic synthesis of qualitative research in systematic reviews. BMC Med Res Methodol 2008;8:45.doi:10.1186/1471-2288-8-45). I’d like it be clearer exactly what and how the authors are doing in terms of synthesis and to see a justification for the choice of synthesis approach.”

We have explicated in greater detail what the thematic synthesis process will entail and have provided justification for using this form of analysis. The data synthesis section now states:

“A thematic synthesis approach as established by Thomas & Harden, will be used to synthesise the findings. Through this approach, four steps will be undertaken to achieve codes, refine themes, and synthesise the data. These four stages are: 1) familiarisation with data through in-depth reading of included papers, 2) line by line coding of the extracted data and critical appraisal results, 3) categorising these codes into descriptive themes and, 4) developing these descriptive themes into analytical ones which provide deeper insights than the original studies¹⁰.

This process will consist of coding the data extracted from the included articles and applying initial codes which are close to the analysed data. This will involve establishing which aspects of hearing loss were investigated, how they were investigated, and how grounded theory was used as a methodology in its applications of central principles such as theoretical sampling, constant comparative analysis and theoretical saturation etc. for investigating the specific hearing loss area. The results of the critical appraisal will then illustrate where grounded theory has been used rigorously and less so, to investigate hearing loss. From descriptive themes, analytical themes will follow, and these will inform a set of guiding statements for researchers wishing to use grounded theory to investigate hearing loss in future studies. Overall, thematic synthesis was chosen as it enables the greater generalisability of results.”

Comment 9. “Also given the focus it would have been helpful to have understood the background of those doing the research, and perhaps to have seen them be reflexive about their positions.”

Thank you for this suggestion. We will certainly include a reflexive section in the systematic review manuscript where the team members will each individually reflect on their own theoretical predispositions, biases, and decisions made during the research process. In this protocol manuscript we have added a brief reference to the background of the research team in the “risk of bias” section:

“Within the research team, two are qualitative experts (NW & DC), with the remaining four members having experience of conducting qualitative research investigating aspects of hearing loss (HH, DH, & YA). DC and YA have experience in the application of grounded theory methodologies. To avoid bias

during the conduction of this systematic review, each author will reflect on their theoretical predispositions, biases, decisions made during the research process and any problems faced. Risk of bias during analysis will be avoided by assessing studies using the GUREGT guidelines, the GRADE-CERQual tool to assess confidence in the data synthesis, and by considering the funding and conflict(s) of interest statements outlined by each study during data extraction to evaluate whether selective reporting of outcomes was present.”

Comment 10: “The authors make no reference to a scoping search and I wondered whether they have done this and got a sense of how many studies they are potentially looking at and where the issues might be in terms of inclusion/exclusion.”

A scoping search was completed. The following paragraph has been added under the ‘information sources’ section.

“To check the feasibility of this systematic review, a scoping search was conducted across multiple databases including Medline, PubMed and PsycINFO. This indicated that there were sufficient relevant papers, approximately 30, to perform a useful synthesis. The searches also informed the key terms for use in the formal search strategy.”

Comment 11: “I’m not sure what the “grounded theory” keywords encompass but wondered whether additional textword searches e.g. “constant comparison” “theoretical sampling” might identify research that might otherwise be missed. Or look for papers which cite key GT texts?”

Thank you for raising this point. Based on our initial scoping searches this was not a concern. In this systematic review we focus on studies that have reported grounded theory principles, to the degree that they explicitly use the term grounded theory within their methodological approach. In addition, these studies will be required to include a peer reviewed reference for their adopted grounded theory approach. For greater clarification, the following was added to the Eligibility Criteria section: ELIGIBILITY CRITERIA: “Studies included in this review will have used grounded theory methodology with an appropriate peer-reviewed reference for that adopted approach. For a study to be included, it can apply grounded theory as an overarching methodology, or be embedded into it amongst other qualitative methodologies such as case studies, ethnography, narratives, and phenomenology. During full text screening, if a study explicitly states that grounded theory was used during data collection and/ or analysis, then they will be eligible for inclusion. Studies that do not explicitly state use of grounded theory methodology will be excluded...”

Comment 12: “I wasn’t sure what the limits are for the research in grounded theory. There are many people who take aspects of grounded theory as part of their qualitative approach but who do not “claim” to be using grounded theory. I conducted a review of qualitative research in ENT and found that the underlying approach was often unclear, particularly when reporting in journals with a shorter word limit or where the audience might be less familiar with different qualitative approaches.”

Thank you again for raising this concern, it is indeed one the team have discussed in detail. As you have stated, many researchers do not “claim” to be using grounded theory, and simply imply that their methodology is informed by a grounded theory approach. But for this systematic review we are only including studies that explicitly state that they are using grounded theory methodology. For clarity, please refer to comment 11.

Comment 13: “The authors say that “The data collected in those included studies will include qualitative primary data occurring in the form of transcripts or quotes from audio interviews with participants, patient journals, written reflections of patients, family members of PHL, or audiologists, memos of progression of study, initial themes and analysis, and observational” (this sentence appears not to be complete). Here, and elsewhere, the authors only refer to audiologists rather than other

health professionals – I wasn't sure whether the authors are interested in other health professionals who are involved in the care of people with hearing loss and their perspectives and interactions with people with hearing loss as well?"

Thank you for highlighting this point. The research team have changed this to include all professionals within audiology. Therefore, we have added the following:

"The data collected in those included studies will include qualitative primary data occurring in the form of transcripts or quotes from audio interviews with participants, patient journals, and written reflections of patients, family members of PHL or professionals working within audiology, memos of progression of study, initial themes, analyses, and observational notes."

Comment 14: "There is some variation in terminology for hearing loss. Terminology from mild to profound is used in some places but elsewhere the authors refer to "people with no residual hearing" (as an exclusion criteria). Their argument is that the experience of total deafness is different from that of hearing loss e.g. in terms of identity – this may be true but it is likely that there is a continuum with the experience of profound hearing loss but some hearing very similar to that of having no residual hearing at all, (and that other factors are also relevant such as age at which hearing was lost and whether other people in their family have hearing loss). This is partly about how the authors will handle exclusion criteria in practice (if papers refer to profound hearing loss for example and don't explicitly refer to whether residual hearing is present) but this is also somewhere that PPI might have been helpful in helping the authors to think through and justify their focus. This is also somewhere that a scoping search might have been helpful. Linked to the previous point – what will the authors do where the sample is mixed and includes some people with no residual hearing?"

The terminology used for hearing loss has been refined to clearly refer to those with residual hearing only. This can be seen in the participants section below. Our systematic review is concerned with the use of grounded theory within a specific population, people with mild to profound hearing loss (regardless of whether this is acquired or congenital). Our rationale for excluding people with no residual hearing (Deaf) was precisely the reason you provide, in that individuals with no residual hearing often identify differently (and therefore may have very different experiences) to people with some residual hearing. By limiting our methodological examination to studies reporting on a pre-defined population, we can focus solely on the methodological rigour in which Grounded Theory was applied. Where there was a mixed population, we only included studies where the individuals with hearing loss were reported separately. This has been clarified in the Participants section:

"Participants will include people with mild to profound, acquired or congenital, hearing loss. Studies involving individuals with no residual hearing will be excluded. This is because people with no residual hearing often identify differently (and have very different experiences) to people with some residual hearing^{58,59,60}. Studies involving both Deaf and hearing loss participants may ask each individual which group they identify with more and classify them on that basis. These studies will be eligible for inclusion as distinctions between the two groups can be made. Studies that do not specify which participants are Deaf and which have hearing loss will be excluded. "

Comment 15: "I couldn't see anything about exclusion criteria around language. Or about years searched (the authors could exclude anything published prior to 1967)."

The following have been added to the inclusion criteria of studies:

"Only studies published in English will be included. Any studies published prior to 1967 (when Grounded theory methodology was first introduced to the literature; Glaser and Strauss, 1967²⁸) will be excluded."

Comment 16: The authors say that "A review of all titles and abstract resulting from the searches will be performed by YA. During this screening stage, 20% of titles/abstracts will be independently

reviewed by a second author. Any differences will be resolved by a third author. Full text screening will be carried out by YA and 20% of full texts will be independently screened by a second author. Any differences between the two authors will be resolved by a third author”

“It is unclear exactly what (how will the 20% be selected?), when, and why these 20% samples are being undertaken. It might be best to do at least some of this early on in each of the processes to identify any areas of discrepancy, resolve them, and make adjustments to e.g. the inclusion criteria – if the authors wait until they are half way through to carry out this activity then if a problem is identified then a large amount of re-screening would need to be done.”

Thank you for this point. As per comment 3, 100% of the articles will be reviewed by both YA and a second author at every stage of article identification. Full data synthesis will be performed by YA and 100% will be completed independently by a second author. A third author will also independently compare both analyses, provide feedback to YA and the second author, and solve any outstanding disputes. To clarify this the following was added in the data synthesis section:

“The data synthesis will be 100% completed by YA and 100% by a second author, ensuring inter-rater reliability as required for qualitative rigour^{14,15}. Results will be compared and discussed in meetings between both authors. A third author will also independently compare both analyses, provide feedback and solve any outstanding disputes. The GRADE-CERQual (‘Confidence in the Evidence from Reviews of Qualitative research’) ¹⁶ tool will be used to assess the confidence of the data Synthesis findings.”

Comment 17: “I’m not sure that doing methodology research is reason enough not to have a PPI element. Patients might have a view on what the boundaries of the research should be (e.g. around which health professionals are important in supporting them with their hearing loss) or a view on data that it would be interesting to extract (although the focus is on grounded theory the authors do plan to synthesise data on experiences of hearing loss and these findings may be of interest to patients). Patient groups might also be able to help with dissemination. It doesn’t need to be a large element.”

The focus of this review is on the use and rigour of grounded theory methods. We will not review/synthesise the study findings per se, only in the context of critiquing the adopted grounded theory methodologies. Finally, we greatly value your comments regarding the scope of PPI for this research and have made changes accordingly. Please see the Editor’s third request- to see how PPI will now be involved in the systematic review.

VERSION 2 – REVIEW

REVIEWER	Dr. Katherine Allan St. Michael's Hospital, Toronto ON Canada
REVIEW RETURNED	03-Dec-2019

GENERAL COMMENTS	It is difficult to assess if the authors addressed all of the reviewer concerns since their was no response to reviewers document summarizing all of the changes. I had to read through the entire submission to figure out what was changed. In future, please summarize all of the proposed changes in a line by line response to each reviewer comment so that reviewers can clearly see what you changed. The manuscript is much improved. I have one small suggestion - I noticed that for data extraction, you did not address my comment,
--

	that you should have 2 authors independently extract data as opposed to one extracting it and then another one correcting it. This is not good practice and you likely will be criticized for this if you choose this method. Otherwise I have no further comments.
--	---

REVIEWER	Nikki Rousseau University of Leeds, UK
REVIEW RETURNED	16-Jan-2020

GENERAL COMMENTS	Comment 7. I still find this sentence slightly odd: “Included studies can be qualitative or mixed methods research of any design/ methodology; retrospective or prospective studies, before and after comparison studies, RCTs, non-RCTs, cohort studies, prospective observational studies, case-control studies, cross-sectional studies, longitudinal studies. Purely quantitative studies, studies that have not applied grounded theory methodology, articles reporting expert opinions, case reports, practice guidelines, case series, conference abstracts, and book chapters will be excluded.” The authors give a very comprehensive list of designs but these mainly use the language of quantitative research – RCTs etc. rather than qualitative research. Although qualitative research could use a case-control design – its unusual to frame qualitative research in these terms. Comment 8 Thank you for adding the extra detail and for correcting the reference for thematic synthesis. However I’m afraid I’m more confused than I was before by the section on thematic synthesis. This approach doesn’t seem to fit with what they want to do which seems like a very structured data collection and not what I would think of as thematic synthesis. I think this still needs a bit more thought. Comment 13 The authors state that they have clarified that the papers could include other health professionals not only audiologists. But this is not reflected in the text. “Studies included will have adult participants (≥18 years) who are either people with an acquired hearing loss, their family and friends (communication partners), or audiologists.” (page 2 line 33 abstract) Where the authors have clarified the focus it is to other health professionals within audiology. This still needs to be justified – because presumably other health professionals in other settings (GPs – at least for an initial referral) and ENT (if the person with hearing loss is suitable for a cochlear implant for example) could be involved in care. I saw a friend recently with hearing loss who has been almost entirely managed within primary care to date (a couple of years) – by focusing on audiology only the authors could be missing out on important parts of the hearing loss experience – e.g. delays in diagnosis and the impact of this. This relates also to my continuing concerns about PPI (comment 17) Comment 17 – The authors have now added some PPI to support the preparation of a lay summary of review findings. However this does not fully address my concerns. I would expect that people with hearing loss might also have a useful contribution to make on the plans for the research – the protocol etc. While people with hearing loss might not be interested in the different traditions within grounded theory they may have a useful view on for
--

	example, (and as previously flagged), the boundaries to the research e.g. in terms of the health professionals that they see as important to their hearing loss experience. Additional comments The authors make some broad statements: “Grounded theory is the only qualitative methodology that enables the construction of novel theories that are completely emergent from within the data, and comprehensively explore and explain phenomena”. Can theories ever be “completely emergent from within the data”? (I know discussion/arguments about this are as old as grounded theory). Is it really not possible to develop a comprehensive theory to explain phenomena without using grounded theory? The authors cite Cresswell’s work on 5 qualitative approaches – although this is a useful classification, qualitative researchers do not always stick rigidly to one approach – and instead tend to draw on bits from different traditions. A strength of qualitative research is often its recognition and understanding of nuance and I would have liked to see a little more of it here. I noticed some typos e.g.: Page 4 line 22: Too many “also”s The impact of hearing loss is not only psychological, as it can also impact interpersonal relationships of PHL also.
--	--

VERSION 2 – AUTHOR RESPONSE

Reviewer 1.

Comment 1: The manuscript is much improved. I have one small suggestion - I noticed that for data extraction, you did not address my comment, that you should have 2 authors independently extract data as opposed to one extracting it and then another one correcting it. This is not good practice and you likely will be criticized for this if you choose this method. Otherwise I have no further comments.

The authors fully take on board Reviewer one’s comment and have implemented it within the revised manuscript. The following line was added to make this clearer: “YA will extract data for all studies and 100% of the studies will be extracted by a second researcher(s) in line with published guidance.” This was also made explicit in the systematic review flowchart: “Extraction of data (100% by YA; 100% by second author).”

Reviewer 2.

Comment 2: The authors give a very comprehensive list of designs but these mainly use the language of quantitative research – RCTs etc. rather than qualitative research. Although qualitative research could use a case-control design – its unusual to frame qualitative research in these terms.

To avoid confusion, we have removed the listing of quantitative research method designs.

Comment 3: Thank you for adding the extra detail and for correcting the reference for thematic synthesis. However, I’m afraid I’m more confused than I was before by the section on thematic

synthesis. This approach doesn't seem to fit with what they want to do which seems like a very structured data collection and not what I would think of as thematic synthesis. I think this still needs a bit more thought.

Thank you for your comment. We acknowledge that our systematic review is unorthodox due to its critical and methodological basis. We also acknowledge that structured data collection will take place to acquire the relevant data for analysis. However, we aim then to apply thematic synthesis for data analysis, through thematic coding, the formation of descriptive themes, and after refinement, analytical themes which directly answer the review's questions will be established. This process follows the steps identified and applied by Thomas and Harden in their 2008 paper. Therefore, we believe that the thematic synthesis plan outlined adequately facilitates fulfilling the review's aims.

Comment 4:

The authors state that they have clarified that the papers could include other health professionals not only audiologists. But this is not reflected in the text. "Studies included will have adult participants (≥ 18 years) who are either people with an acquired hearing loss, their family and friends (communication partners), or audiologists." (page 2 line 33 abstract)

Thank you, this was an omission. The following has been added in the abstract: "Studies included will have adult participants (≥ 18 years) who are either people with an acquired hearing loss, their family and friends (communication partners), or healthcare practitioners including audiologists, GPs, ENT specialists, and hearing therapists."

Comment 5: Where the authors have clarified the focus it is to other health professionals within audiology. This still needs to be justified – because presumably other health professionals in other settings (GPs – at least for an initial referral) and ENT (if the person with hearing loss is suitable for a cochlear implant for example) could be involved in care. I saw a friend recently with hearing loss who has been almost entirely managed within primary care to date (a couple of years) – by focusing on audiology only the authors could be missing out on important parts of the hearing loss experience – e.g. delays in diagnosis and the impact of this. This relates also to my continuing concerns about PPI (comment 6)

Thank you. Following PPI in reviewing the protocol we have extended the review to include additional health professionals (as per response to Comment 4).

Comment 6: The authors have now added some PPI to support the preparation of a lay summary of review findings. However this does not fully address my concerns. I would expect that people with hearing loss might also have a useful contribution to make on the plans for the research – the protocol etc. While people with hearing loss might not be interested in the different traditions within grounded theory they may have a useful view on for example, (and as previously flagged), the boundaries to the research e.g. in terms of the health professionals that they see as important to their hearing loss experience.

Thank you, we agree. We have added the following in the patient and public involvement section: “People with hearing loss were consulted in the identification of the most relevant healthcare practitioner roles when seeking hearing healthcare services. The prominent professions identified were audiologists, GPs, ENT specialists, and hearing therapists. Therefore, they have been included within the inclusion criteria.”

Comment 7: The authors make some broad statements: “Grounded theory is the only qualitative methodology that enables the construction of novel theories that are completely emergent from within the data, and comprehensively explore and explain phenomena”. Can theories ever be “completely emergent from within the data”? (I know discussion/arguments about this are as old as grounded theory). Is it really not possible to develop a comprehensive theory to explain phenomena without using grounded theory? The authors cite Cresswell’s work on 5 qualitative approaches – although this is a useful classification, qualitative researchers do not always stick rigidly to one approach – and instead tend to draw on bits from different traditions. A strength of qualitative research is often its recognition and understanding of nuance and I would have liked to see a little more of it here.

Thank you very much for this comment. The authors acknowledge that there is a fluidity in the application of qualitative methods and that this flexibility is what can establish unprecedented insights. Grounded theory is the qualitative methodology specifically dedicated to forming new theories. Therefore, we’ve removed the word only, acknowledging that the establishment of new theories is possible using other qualitative methodologies, whilst still highlighting grounded theory’s main function for doing so. This now reads on page 5: “Grounded theory is a qualitative methodology developed specifically to enable the construction of novel theories that are directly emergent from within the data, which comprehensively explore and explain phenomena.”

Comment 8: I noticed some typos e.g.: Page 4 line 22: Too many “also”s The impact of hearing loss is not only psychological, as it can also impact interpersonal relationships of PHL also.

Thank you, the manuscript has now been revised and typos corrected.